# Risk relationship between inflammatory bowel disease and urolithiasis: A two-sample Mendelian randomization study

**Wenqiang Fu[1][◎], Bin Zhu[2][◎], Jun Chen[3], Xuelin Jin[1]***

**1** Affiliated Hospital, Anorectal, Panzhihua University, Panzhihua, Sichuan, China, **2** Outpatient Department, Tibet Military Region General Hospital of PLA, Lhasa, China, **3** Jiangxi University of Traditional Chinese Medicine, Nanchang, Jiangxi, China

◎ These authors contributed equally to this work.
* 1934316454@qq.com

## Abstract

### Background

The causal genetic relationship between common parenteral manifestations of inflammatory bowel disease (IBD) and urolithiasis remains unclear because their timing is difficult to determine. This study investigated the causal genetic association between IBD and urolithiasis using Mendelian randomization (MR) based on data from large population-based genome-wide association studies (GWASs).

### Methods

A two-sample MR analysis was performed to assess the potential relationship between IBD and urolithiasis. Specific single nucleotide polymorphism data were obtained from GWASs, including IBD (n = 59957) and its main subtypes, Crohn's disease (CD) (n = 40266) and ulcerative colitis (UC) (n = 45975). Summarized data on urolithiasis (n = 218792) were obtained from different GWAS studies. A random-effects model was analyzed using inverse-variance weighting, MR-Egger, and weighted medians.

### Results

Genetic predisposition to IBD and the risk of urolithiasis were significantly associated [odds ratio (OR), 1.04 (95% confidence interval [CI], 1.00–.08), $P = 0.01$]. Consistently, the weighted median method yielded similar results [OR, 1.06 (95% CI, 1.00–1.12), $P = 0.02$]. The MR-Egger method also demonstrated comparable findings [OR, 1.02 (95% CI, 0.96–1.08), $P = 0.45$]. Both funnel plots and MR-Egger intercepts indicated no directional pleiotropic effects between IBD and urolithiasis. CD was strongly associated with it in its subtype analysis [OR, 1.04 (95% CI, 1.01–1.07), $P = 0.01$], and UC was also causally associated with urolithiasis, although the association was not significant [OR, 0.99 (95% CI, 0.95–1.03), $P = 0.71$].

**Data Availability Statement:** Genetic data pertaining to IBD, CD and UC, as well as urinary stone disease, were obtained from the GWAS (https://gwas.mrcieu.ac.uk/datasets/). The datasets are ebi-a-GCST004131, ebi-a-GCST004132 and

ebi-a-GCST004133 corresponding to IBD,CD and UC, respectively. The FINNGEN database was employed to gather the GWAS summary data for the investigation of urolithiasis, derived from the following link (https://gwas.mrcieu.ac.uk/datasets/finn-b-N14_UROLITHIASIS/).

**Funding:** This work was supported by Key Research and Development Program of Tibet Autonomous Region (Grant No. XZ202001ZY0059G to Bin Zhu). The funders had no role in the study design, data collection, or analysis but provided the initial writing of the manuscript and the supervision, review, and editorial advice of subsequent articles. In addition to this, "There was no additional external funding received for this study".

**Competing interests:** The authors have declared that no competing interests exist.

## Conclusion

A unidirectional positive causal correlation was identified between IBD and urolithiasis, with varying degrees of association observed among the different subtypes of IBD. Recognizing the increased incidence of urolithiasis in patients with IBD is crucial in clinical practice. Early detection and surveillance of IBD, improved patient awareness, adoption of preventive strategies, and promotion of collaborative efforts among healthcare providers regarding treatment methodologies are vital for improving patient outcomes.

## 1 Introduction

Urolithiasis, a prevalent condition evaluated and managed in the emergency department, involves the formation of calculi in the kidneys, ureters, bladder, or urethra [1]. Epidemiological data have indicated that the symptomatic presentation of urolithiasis is of great significance, with few patients developing symptoms after the age of 70 years, whereas >70% of patients with urolithiasis are between the ages of 20 and 50 years [2]. The clinical presentation of urolithiasis exhibits remarkable diversity, contingent on the precise location, size, and propensity for obstruction by the calculi. Notable clinical indicators of urolithiasis include renal colic, hematuria, abdominal pain, and urinary tract infection. In addition, a wide range of complications has been reported, including hydronephrosis, pyelonephritis, and urinary tract obstruction. These deleterious consequences can ultimately culminate in severe sequelae, such as sepsis and mortality attributed to obstructive infections [3, 4]. These grave complications pose a substantial threat to the patients' quality of life, necessitating prompt and effective intervention [5]. Moreover, emerging research has revealed a possible link between urolithiasis onset and inflammatory bowel disease (IBD), demonstrating a previously unrecognized association between these distinct pathologies [6].

IBD is a chronic inflammatory disorder that affects the gastrointestinal tract and has two distinct subtypes: Crohn's disease (CD) and ulcerative colitis (UC) [7]. Patients with this condition often present with multifocal lesions ranging from the oral cavity to the anus, accompanied with various symptoms attributed to both local and systemic inflammations stemming from the disease. A growing body of evidence has suggested an association between IBD and the increasing incidence of urolithiasis [8, 9]. The connection between IBD and kidney stones was first elucidated by Gelzayd et al. in 1968, marking a significant milestone in understanding the interplay between these two conditions [10]. Subsequently, a Swiss cohort study involving 2323 individuals has reported that 4.6% of patients with CD and 3.0% of patients with UC developed urolithiasis. However, a direct comparison with a non-IBD population was not performed in this study [11]. Interestingly, in Scandinavia, 15% of urolithiasis cases have been attributed to metabolic, infectious, or anatomical/functional factors, a result that is particularly striking in patients diagnosed with IBD [12]. Moreover, patients with both IBD and urolithiasis are at an increased risk of developing urinary tract infections, renal impairment, and sepsis compared with individuals with urolithiasis alone [13]. The association of inflammatory IBD and renal involvement is well known; the most frequent renal diseases are nephrolithiasis, tubulointerstitial nephritis, glomerulonephritis and amyloidosis. However, most studies exploring the link between IBD and urolithiasis are either dated or based on relatively small or selective patient cohorts. Furthermore, clinical observational studies inherently carry a risk of bias, such as statistical confounders or environmental exposures. Thus, the role of IBD in the development of urolithiasis needs to be investigated.

To comprehensively explore the intricate causal relationship between IBD and urolithiasis, including the strength of the association and directionality of causation, we employed a powerful analytical approach known as Mendelian randomization (MR). Originally proposed by Katan in 1986, this innovative methodology leverages the concept of genetic variation as an instrumental variable (IV) to assess the causal relationship between exposure and outcomes [14, 15]. By capitalizing on the random distribution of genetic variation, MR effectively mitigates the perils of confounding factors and reverse causality, emulating the randomization process of randomized controlled trials (RCTs) and circumventing the confounding effects and potential biases encountered in conventional RCTs [16].

## 2 Materials and methods

### 2.1 Study design

In this study, IBD was used as an exposure factor, single nucleotide polymorphisms (SNPs) significantly associated with IBD were used as IVs, and urolithiasis was used as the outcome variable. A two-sample MR analysis was performed to analyze causal association. Cochran's Q test was used to conduct a heterogeneity test, and sensitivity analysis was conducted to verify the reliability of the obtained results.

To select suitable IVs for the two-sample MR analysis, this study adopted the following three key assumptions: (1) IVs and IBD are significantly correlated, (2) IVs are not associated with all confounders associated with IBD-urolithiasis, and (3) IVs can only affect the outcome through their association with IBD [17]. The experimental design process is illustrated in Fig 1.

### 2.2 Data source

Genetic data pertaining to IBD, CD, and UC, as well as urinary stone disease, were obtained from the authoritative website (https://gwas.mrcieu.ac.uk). To mitigate the potential influence of race-related confounding factors, the genetic background of the study population predominantly comprised of individuals of European descent. The comprehensive dataset employed in our study encompassed a substantial cohort of 25,042 IBD cases and 34,915 controls; 12,194 CD cases and 28,072 controls; and 12,366 UC cases and 33,609 controls. Specifically, the genetic information pertinent to urolithiasis originated from the FinnGen Biobank, which encompasses a cohort of 5,347 urolithiasis cases and 213445 controls, comprising both men and women, and an impressive number of SNPs, totaling to 1,6380,466. As all the utilized data

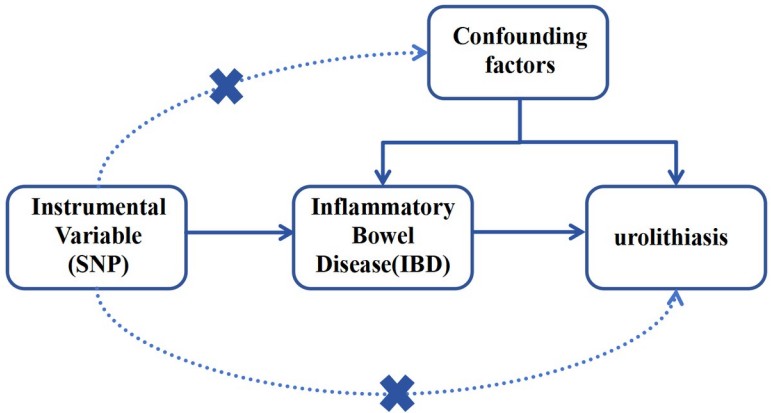

**Fig 1. Plot of key assumptions for Mendelian randomization analysis.**

are publicly available and the information is derived from existing sources, no additional ethical approval was required for our study.

## 2.3 Selection criteria for genetic variants

In our study, to effectively deal with the issue of genetic variants having minimal correlation with exposure factors, often labeled as "weak instruments," we adopted rigorous standards for selecting SNPs with a strong link to IBD. Initially, we filtered a collection of SNPs that exceeded the significance level of $P < 5 \times 10^{-6}$ utilizing a method based on spatial proximity. For SNPs directly relevant to IBD, we imposed an even stricter criterion, demanding a P-value of $< 5 \times 10^{-8}$. The F-statistic was calculated in the regression analysis to counteract potential distortions caused by weak instruments. Typically, an F-statistic of $>10$ indicates the absence of weak instruments, as per the findings in reference [18].

Regarding our second hypothesis, a genetic variant falls under the category of pleiotropy if it demonstrates unobserved confounding effects on its links to outcomes, thereby possibly breaching the principles of independence and exclusivity. To address this challenge, we utilized various techniques such as inverse-variance weighting (IVW) [19], MR-Egger [20], and weighted median (WM) [21]. These approaches facilitate the creation of regression models that consider the relationship between gene exposure and clinical outcomes. This allowed for the examination and rectification of skewness caused by the pleiotropic influence of the IVs.

We addressed the concept of linkage disequilibrium (LD), which indicates the propensity for genetic variants in close proximity to be inherited together. To ensure that our IVs were independent and minimize the impact of LD, we set specific criteria for the LD of IVs. These criteria include a genetic separation of at least 10,000 kilobases (kb) and an $r^2$ value of $<0.001$. We then selected significant SNPs associated with exposure factors from genome-wide association studies data related to outcome variables. Comprehensive details such as the identity of the IVs, effector alleles, size of the allele effect (β), standard error (SE), and P-values were all carefully documented.

To eliminate false correlations arising from variations in genetic ancestry, we deliberately included European heritage populations that shared comparable genetic profiles. This strategy was effectively implemented to bypass disparities that often result from population stratification.

## 2.4 Mendelian randomization analysis

In this research, we utilized the "TwoSampleMR" package in R version 4.1.2 to perform an MR analysis. The initial phase involved evaluating the relationship between each SNP and determining its Wald ratio. Subsequently, Wald ratios were combined using the IVW method to evaluate the relationship between IBD and urolithiasis. We employed MR-Egger regression and WM methods to validate and assess the robustness of our MR results. The IVW approach is predominantly based on the inverse variance of each IV as the basis for computing their respective weights [22]. This process assumes the validity of all IVs, allowing for the assessment of horizontal pleiotropy [21]. Meanwhile, the MR-Egger regression approach incorporates the reciprocal of the outcome variance as weights in the regression model, while simultaneously including an intercept term [23]. The WM, as the median of the weighted empirical density function defined as the ratio estimate, can still provide a consistent effect estimate even when the proportion of invalid IVs is high at 50%, and the difference of accuracy among the IVs is substantial.

## 2.5 Pleiotropy and sensitivity analysis

Leave-one-out and MR-Egger methods were used to assess the robustness and reliability of the results. The MR-Egger method was used to detect the presence of horizontal pleiotropy, which was indicated by a substantial deviation of the MR-Egger intercept term from zero [24]. By contrast, the leave-one-out method systematically excludes each IV individually and calculates the results using the remaining IVs. When a specific MR result indicates no substantial deviation from the aggregate outcomes, this implies that the involved IV does not exert an unspecific impact on the estimation of the effect.

# 3 Results

## 3.1 Mendelian randomization analysis

In this study, we aimed to investigate the causal relationship between inflammatory bowel disease (and its two subtypes, Crohn's disease and ulcerative colitis) and urolithiasis using a two-sample MR approach. Three sets of genetic instruments were utilized in the analysis, all drawn from publicly available genetic summary data. The first set consisted of 259 single-nucleotide polymorphisms (SNPs) robustly linked to inflammatory bowel disease (IBD), the second set was comprised of 81 SNPs associated with inflammatory bowel disease (CD) and the third set consisted of 175 SNPs associated with ulcerative colitis (UC).

The analysis using the IVW approach indicated a notable link between genetic predisposition to IBD and an increased risk of urolithiasis [odds ratio (OR), 1.04 (95% confidence interval [CI], 1.00–1.08), $P = 0.01$]. In a similar vein, the WM also produced analogous outcomes [OR, 1.06 (95% CI, 1.00–1.12), $P = 0.02$]. Furthermore, the MR-Egger method also corroborated these findings [OR, 1.02 (95% CI, 0.96–1.08), $P = 0.45$]. Importantly, all three methodologies consistently suggested the same causal direction and significant statistical impact (Fig 2).

Additionally, we performed MR studies to explore the link between two variants of IBD (CD and UC) and the likelihood of developing urolithiasis. The IVW method approach indicated a notable association for CD and urolithiasis, with an OR of 1.04 (95% CI, 1.01–1.07) and a P-value of 0.01. Conversely, the outcomes from the WM method [OR, 1.01 (95% CI, 0.96–1.06), $P = 0.56$] and the MR-Egger analysis [OR, 1.05 (95% CI, 0.97–1.13), $P = 0.17$] did not reach statistical significance. By contrast, the IVW results for UC suggested a possible causal link with urolithiasis, but without reaching statistical significance [IVW: OR, 0.99 (95% CI, 0.95–1.03), $P = 0.71$; WM: OR, 0.96 (95% CI, 0.91–1.01), $P = 0.16$; MR-Egger: OR, 0.92 (95% CI, 0.91–1.01), $P = 0.10$]. The comprehensive results are illustrated in Fig 3.

## 3.2 Sensitivity analysis

Cochran's Q tests were conducted to assess heterogeneity among the SNPs closely associated with IBD that were included in our study. Neither the IVW method ($P = 0.31$) nor the MR-Egger regression ($P = 0.29$) indicated significant heterogeneity. To further investigate the possibility of pleiotropic effects, we performed MR-PRESSO. The results of these tests revealed no signs of directional pleiotropy (P = 0.42). Additionally, funnel plot analysis revealed a symmetric distribution of the effects of causal association for each individual SNP-based IV, suggesting that our findings were likely free from bias (Fig 4).

Furthermore, we conducted a "leave-one-out" sensitivity analysis by sequentially excluding each SNP from the analysis. The IVW analysis using the remaining 222 SNPs yielded results similar to those obtained when all SNPs were included. No SNPs had a significant impact on causal association estimates after exclusion (Fig 5).

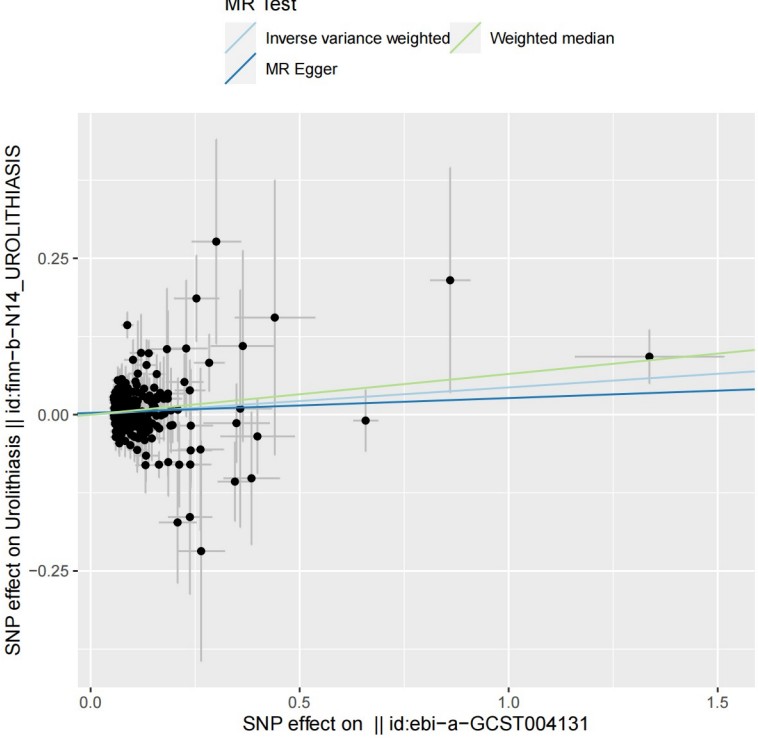

**Fig 2. Scatter plot.** The slope of each line corresponds to the estimated Mendelian randomization effect in different models.

## 4 Discussion

Understanding the etiology of urolithiasis is crucial for its prevention, diagnosis, and treatment. This study represents the first application of MR to assess the causal relationship between IBD and urolithiasis risk. The results of the IVW and WM analyses suggested that genetic factors associated with IBD predict an increased risk of urolithiasis. Various sensitivity analyses confirmed a causal relationship between these two diseases. Additionally, we conducted separate MR analyses for the IBD subtypes and identified a significant causal relationship between CD and urolithiasis. Although UC and urolithiasis may have a causal relationship, the results were not statistically significant.

Previous research has demonstrated an increased risk of urolithiasis in patients with IBD. A nationwide cohort study from Denmark from 1977 to 2018 revealed that the risk of urolithiasis in patients with IBD doubled after diagnosis and increased by 42% before diagnosis [25]. The study also suggested that this increased risk may be related to the severity of IBD itself, surgical interventions, changes in intestinal absorption, and increased risk after antitumor necrosis factor therapy and surgery [25]. Another study from the UK has reported that the incidence rates of urolithiasis and pyelonephritis among patients with CD ranged from 4% to 23%, with a risk 10 to 100 times higher than that in the general population or those with UC [6].

Similarly, an observational study has indicated that certain types of urinary stones are more common in patients with IBD. Hyperoxaluria and hypocitraturia are the main metabolic changes leading to urolithiasis [26]. Stark et al. identified 36,771 cases of urolithiasis among 8,828,522 hospitalized patients with IBD, with 230 cases associated with CD (OR, 1.99; 95%

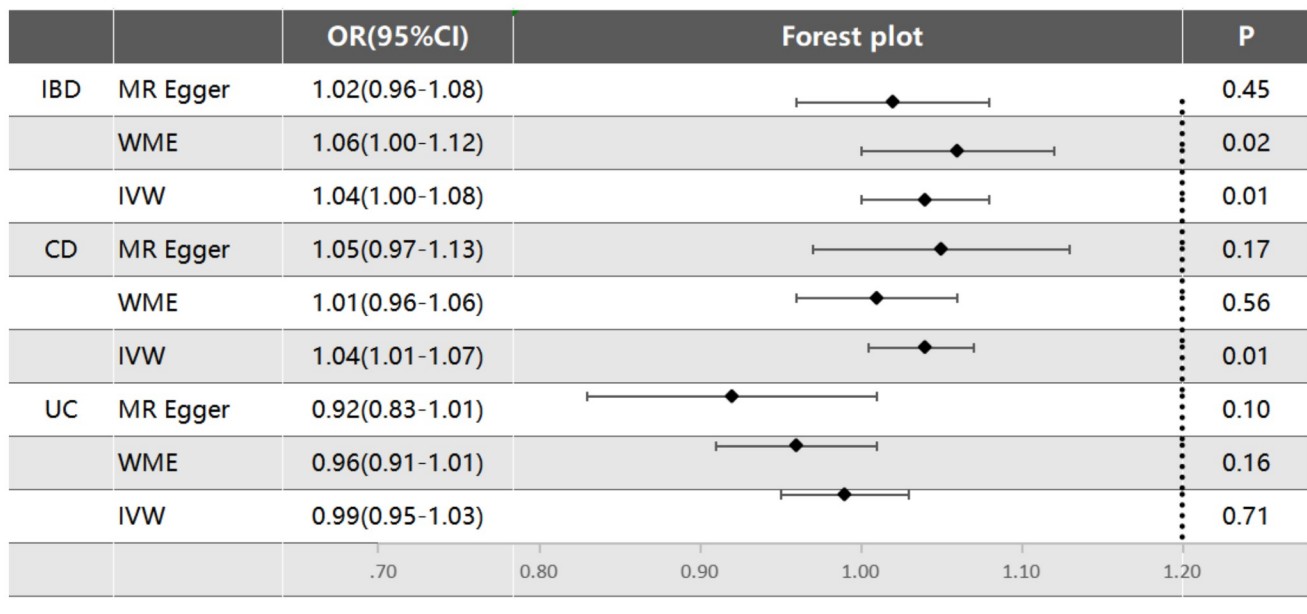

| | | OR(95%CI) | Forest plot | P |
|---|---|---|---|---|
| IBD | MR Egger | 1.02(0.96-1.08) | | 0.45 |
| | WME | 1.06(1.00-1.12) | | 0.02 |
| | IVW | 1.04(1.00-1.08) | | 0.01 |
| CD | MR Egger | 1.05(0.97-1.13) | | 0.17 |
| | WME | 1.01(0.96-1.06) | | 0.56 |
| | IVW | 1.04(1.01-1.07) | | 0.01 |
| UC | MR Egger | 0.92(0.83-1.01) | | 0.10 |
| | WME | 0.96(0.91-1.01) | | 0.16 |
| | IVW | 0.99(0.95-1.03) | | 0.71 |

Forest plot to visualize causal effect of IBD on the risk of Urolithiasis by three methods。 Abbreviations: IVW indicates inverse-variance weighted.

**Fig 3. Forest plot.** Effects of inflammatory bowel disease on urolithiasis.

CI, 1.74–2.27) and 102 cases associated with UC (OR, 1.63; 95% CI, 1.34–1.99) [27]. Consistent with previous findings, our MR study revealed that genetic susceptibility in patients with IBD increased the risk of urolithiasis by approximately 4%.

However, the specific mechanisms underlying the increased risk of urolithiasis in patients with IBD remain unclear. Currently, most researchers believe that this is associated with the following four aspects:

## 4.1 Systemic inflammation and urolithiasis

The chronic inflammatory state observed in IBD can lead to systemic changes that promote urolithiasis [28]. Elevated levels of pro-inflammatory cytokines, such as interleukin-6 and tumor necrosis factor-alpha, are involved in stone formation by regulating urinary calcium excretion and promoting the deposition of calcium oxalate crystals in the kidneys [29]. Furthermore, systemic inflammation can increase the production of reactive oxygen species (ROS), which are highly reactive molecules that cause oxidative damage to cells and tissues. ROS can promote stone formation by altering urine composition and facilitating the crystallization of stone-forming substances such as calcium and oxalate [30].

## 4.2 Intestinal inflammation and urolithiasis

IBD-related intestinal inflammation can disrupt the intestinal barrier, leading to increased intestinal permeability and translocation of bacterial products into the bloodstream [31]. This phenomenon, known as bacterial translocation, triggers systemic inflammation and activates immune responses, thereby contributing to urolithiasis. Inflammation in IBD affects acid–

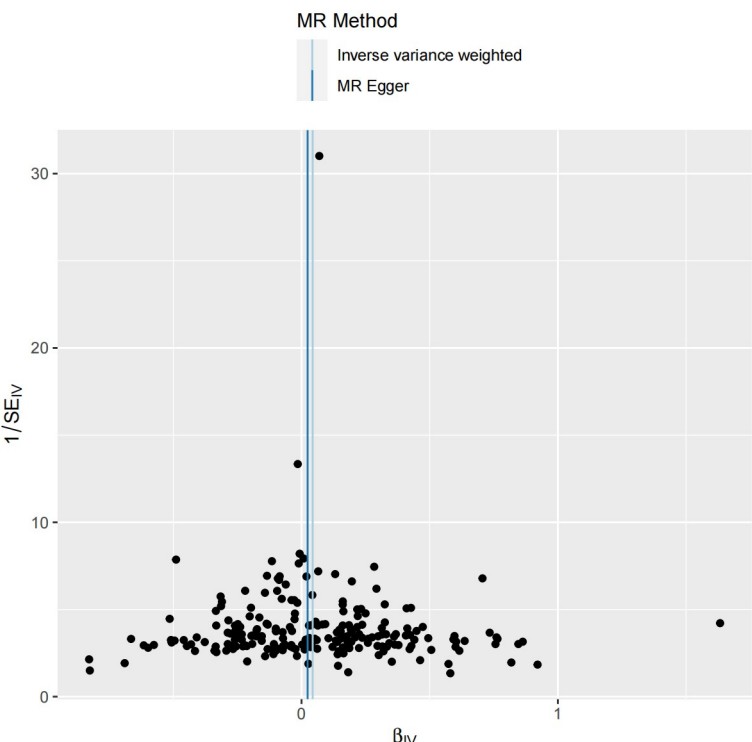

**Fig 4. Funnel plot.** Global heterogeneity in the effect of inflammatory bowel disease on urolithiasis risk assessed by Mendelian randomization.

base balance and electrolyte levels in the body [32]. These changes can alter the urine pH and the concentration of substances involved in stone formation, creating an environment favorable for urinary stone formation.

### 4.3 Gut microbiota alterations and urolithiasis

The gut microbiota plays a crucial role in maintaining immune homeostasis and metabolic balance [33]. Dysbiosis of the gut microbiota has been observed in patients with IBD, and is characterized by reduced microbial diversity and changes in the relative abundance of specific bacterial species [34]. This dysbiotic state can promote urolithiasis by affecting the metabolism of oxalate and other stone-forming components in the urinary tract [35].

### 4.4 Treatment interventions and urolithiasis

Certain interventions used in the management of IBD, such as corticosteroids and diuretics, can further predispose patients to urolithiasis [36]. Additionally, colonic surgery, specifically colectomy, in patients with UC can lead to changes in urine composition and stone formation [37]. The commonly used IBD medication 5-aminosalicylic acid has been implicated in kidney stone formation in relevant case reports [38].

Understanding the increased incidence of urolithiasis in IBD carries several important clinical implications.

For early detection and monitoring, patients with IBD should be closely monitored for urolithiasis [37]. Healthcare providers should be vigilant in assessing urinary symptoms such as hematuria (blood in urine) and abdominal pain, which may indicate the presence of urinary

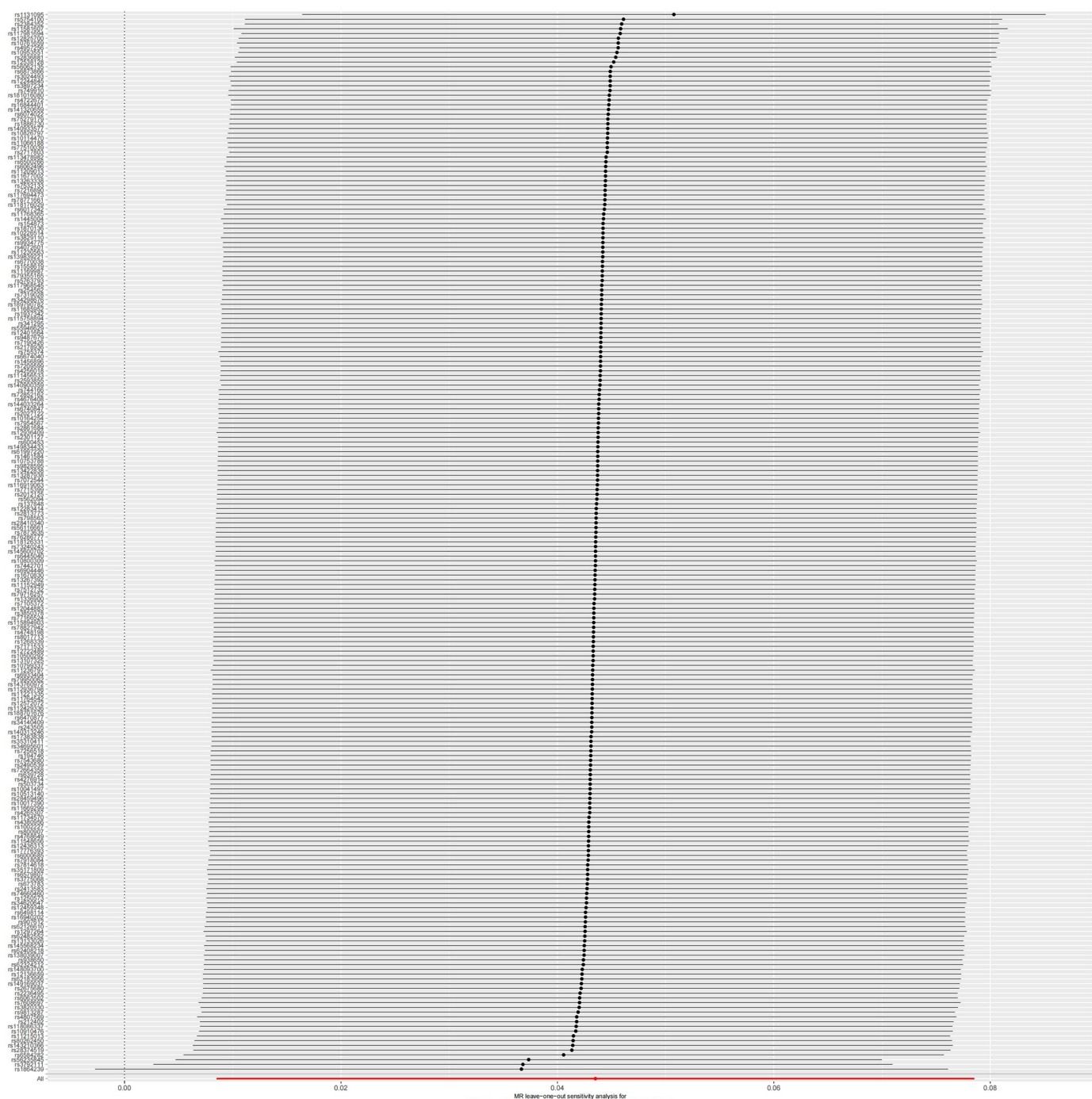

**Fig 5. Leave-one-out plot.** Each black dot in the forest plot represents the Mendelian randomization analysis excluding single nucleotide polymorphisms individually.

stones [39]. Regular monitoring of urinary parameters, including urine pH, calcium, and oxalate levels, can help identify at-risk individuals and enable early intervention.

To improve patient education, healthcare providers should educate patients with IBD on the increased risk of urolithiasis.

Regarding treatment strategies, the management of urolithiasis in patients with IBD may require a multidisciplinary approach involving gastroenterologists, urologists, and nephrologists. Treatment strategies should be individualized considering the underlying IBD and any specific factors contributing to stone formation. Therapeutic interventions may include pharmacological control of inflammation, dietary adjustments, increased fluid intake, and surgical removal of stones in certain cases.

Regarding preventive strategies, considering the increased risk of urolithiasis in patients with IBD, preventive measures should be adopted. These may involve lifestyle modifications such as maintaining adequate hydration, adopting a balanced diet, appropriate intake of calcium and oxalate, and avoidance of known stone-forming foods. Under certain circumstances, medications or supplements may be prescribed to modify the urinary chemistry and reduce the risk of stone formation.

For coordinated care, collaborative efforts among gastroenterologists, urologists, and nephrologists are pivotal in ensuring the comprehensive and coordinated management of patients with both IBD and urolithiasis [40]. This entails regular communication, shared medical records, and shared decision-making to optimize patient outcomes and minimize the overall impact of these conditions on patient well-being.

By understanding the increased incidence of urolithiasis in patients with IBD, healthcare providers can implement appropriate strategies for the detection, management, and prevention of urinary stones in affected individuals, thereby enhancing patient prognosis and quality of life.

## 5 Strengths and limitations

Using MR analysis, we established a positive, unidirectional causal relationship between IBD and urolithiasis in a population of European ancestry. The advantage of MR lies in its ability to evaluate the causal relationship between genetically predicted IBD risk and urolithiasis within the same study population, thereby avoiding biases related to confounding and reverse causality as per Mendel's second law of inheritance.

However, this study had several limitations. First, MR is subject to all the limitations of IV analysis and several limitations specific to its genetic underpinnings, including confounding factors, weak instrument bias, pleiotropy, adaptation, and failure of replication. Moreover, randomization or causality should be eliminated from reports of its use, and MR should be referred to as "genetic IV analysis" [41]. Second, the examined genes were limited to populations of European ancestry, warranting further MR investigations in Asian and African ancestry populations to confirm the causal relationship between IBD and urolithiasis. Although a causal relationship between IBD and urolithiasis has been demonstrated, the underlying mechanism by which IBD increases the risk of urolithiasis requires further research. Finally, this study identified varying causal relationships between different subtypes of IBD and urolithiasis. However, the specific mechanisms remain unclear, and the potential influence of common pathogenic factors, such as smoking and alcohol consumption, cannot be ignored.

## 6 Conclusion

In this extensive MR study, we identified a consistent and positive causal link between IBD and urolithiasis. This association demonstrated different strengths across various IBD subtypes. Therefore, recognizing the increased incidence of urolithiasis in individuals with IBD has crucial clinical implications. Early identification and monitoring of IBD, enhanced patient education, implementation of preventive measures, and fostering inter-physician coordination in treatment approaches are important for optimizing patient outcomes.

## Acknowledgments

We would like to thank Dr. Zhu and Dr. Jin for their contributions to our article, and we also gratefully acknowledge the authors and participants of all GWAS for which we used summary statistics.

## Author Contributions

**Conceptualization:** Wenqiang Fu, Xuelin Jin.

**Data curation:** Wenqiang Fu, Jun Chen.

**Formal analysis:** Wenqiang Fu, Jun Chen.

**Investigation:** Bin Zhu.

**Supervision:** Wenqiang Fu, Bin Zhu, Xuelin Jin.

**Writing – original draft:** Wenqiang Fu, Jun Chen.

**Writing – review & editing:** Bin Zhu, Xuelin Jin.

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
