## [Decision Letter · Decision Letter 0]

9 Nov 2023

PONE-D-23-27679Inflammatory Bowel Disease may Increase the Risk of Urolithiasis: A two-sample Mendelian Randomization studyPLOS ONE

Dear Dr. Jin,

Thank you for submitting your manuscript to PLOS ONE. After careful consideration, we feel that it has merit but does not fully meet PLOS ONE’s publication criteria as it currently stands. Therefore, we invite you to submit a revised version of the manuscript that addresses the points raised during the review process.

We look forward to receiving your revised manuscript.

Kind regards,

Gurinder Kumar, MD

Academic Editor

PLOS ONE

Journal Requirements:

-https://www.researchsquare.com/article/rs-3167883/v1?
https://www.medrxiv.org/content/10.1101/2020.07.09.20150136v1.full?
https://www.hindawi.com/journals/dth/2023/1353823/?

3. In your revision ensure you cite all your sources (including your own works), and quote or rephrase any duplicated text outside the methods section. Further consideration is dependent on these concerns being addressed.

4. We suggest you thoroughly copyedit your manuscript for language usage, spelling, and grammar. If you do not know anyone who can help you do this, you may wish to consider employing a professional scientific editing service. 

Whilst you may use any professional scientific editing service of your choice, PLOS has partnered with both American Journal Experts (AJE) and Editage to provide discounted services to PLOS authors. Both organizations have experience helping authors meet PLOS guidelines and can provide language editing, translation, manuscript formatting, and figure formatting to ensure your manuscript meets our submission guidelines. To take advantage of our partnership with AJE, visit the AJE website (http://aje.com/go/plos) for a 15% discount off AJE services. To take advantage of our partnership with Editage, visit the Editage website (www.editage.com) and enter referral code PLOSEDIT for a 15% discount off Editage services. If the PLOS editorial team finds any language issues in text that either AJE or Editage has edited, the service provider will re-edit the text for free.

5. Thank you for stating in your Funding Statement: 

This work was supported by Key Research and Development Program of Tibet Autonomous Region (Grant No.XZ202001ZY0059G to Bin Zhu). The funders had no role in the study design, data collection and analysis or preparation of the manuscript, but provided advice on the supervision of the articles and on the review and editing.

Additional Editor Comments:

The authors have used comprehensive Mendelian randomization analysis to assess the potential relationship between IBD and urolithiasis.

Please use consistent abbreviations such as weighted median( WM ) instead of WME.

Reviewers' comments:

Reviewer's Responses to Questions

**Comments to the Author**

1. Is the manuscript technically sound, and do the data support the conclusions?

Reviewer #1: Yes

Reviewer #2: Partly

2. Has the statistical analysis been performed appropriately and rigorously? 

Reviewer #1: I Don't Know

Reviewer #2: I Don't Know

3. Have the authors made all data underlying the findings in their manuscript fully available?

Reviewer #1: Yes

Reviewer #2: Yes

4. Is the manuscript presented in an intelligible fashion and written in standard English?

Reviewer #1: Yes

Reviewer #2: Yes

5. Review Comments to the Author

Reviewer #1: I have limited knowledge of the statistical methods used. With this limitation, my comments are:

1. I do not understand what the three study groups (with their individual controls) are: apart from Crohn's disease and ulcerative colitis, there is a third group of "IBD". What is this third group?

2. The OR is just above 1.00 in these cases. If I understand right, this means the risk of coexistence is only slightly above expected what is expected from the general population.

Minor comments:

3. The Introduction is too lengthy and simplistic. This can be made more precise and focused

4. The authors use a lot of introductory phrases and adjectives, which can be deleted during editing

Reviewer #2: 1.The association of inflammatory Intestinal Bowel disease and renal involvement is well known – the most frequent renal diseases are: nephrolithiasis, tubulointerstial nephritis, glomerulonephritis and amyloidosis.

As an example, PubMED for “renal lithiasis and inflammatory bowel disease” retrieved 147 results.

2. The authors used the Mendelian Randomization Analysis. This method has been criticized as follows:

“In recent years, epidemiologists have increasingly sought to employ genetic data to identify ‘causal’ relationships between exposures of interest and various endpoints – an instrumental variable approach sometimes termed Mendelian randomization.

However, this approach is subject to all of the limitations of instrumental variable analysis and to several limitations specific to its genetic underpinnings, including confounding, weak instrument bias, pleiotropy, adaptation, and failure of replication.

Although the approach enjoys some utility in testing the etiological role of discrete biochemical pathways, like folate metabolism, this method must be treated with all the circumspection that should accompany all forms of observational epidemiology.

Going forward, we urge the elimination of randomization or causality in reports of its use and suggest that Mendelian randomization instead be termed exactly what it is – genetic instrumental variable analysis”.

(Kenneth J Mukamai, Meir J.Stampfer and Eric B.Rimm; European Journal of Epidemiology, volume 35, pages 93-97 (2020).

I suggest that this article should be evaluated also by a Statistics Expert.

6. PLOS authors have the option to publish the peer review history of their article (what does this mean?). If published, this will include your full peer review and any attached files.

Reviewer #1: No

Reviewer #2: **Yes: **Teresa Adragao or Adragao T

---

## [Author Response · Author response to Decision Letter 0]

27 Dec 2023

Dear Editor,

We feel great thanks for your professional review work on our article. As you are concerned, there are several problems that need to be addressed. According to your suggestions, we have made extensive corrections to our previous draft, the detailed corrections are listed below:

1.We have tried our best to modify the format of the article in accordance with the style requirements of PLOS ONE's, which is mainly reflected in terms of titles at all levels, references and file naming requirements, etc.

2.Thank you for your reminding. We have made some improvements such as text weight reduction and sentence modification for repeated text. Specific changes in the “Revised Manuscript with Track Changes” are highlighted in yellow.

3.Thanks for your reminding, we have reviewed and modified the quoted source once again. At the same time, we have also rephrase duplicated text outside the methods.

4.We have corrected as much as possible the language use, spelling and grammar problems that may have appeared in the previous manuscript.

5.Thank you for your suggestions, and changes specific to the funding statement will be attached to the resubmitted cover letter.

6.We have carefully reviewed and revised the references in the article, and the specific modifications are as follows:

1)The number of authors shown in references was revised according to the style requirements of PLOS ONE. (references 5-6, 8-9,11,13,16,18,19,22,28,39,40)

2)For references 12 and 36, considering the published year too long possible adaptation degree is not high, we replaced it.

3)Controversial or inappropriate references such as 24,26,31 have been replaced to better fit the quotation.

The above modifications are reflected in the file "Revised Manuscript with Track Changes".

Dear reviewer,

We would also like to thank all reviewers for their valuable feedback that we have used to improve the quality of our manuscript. The reviewer comments are laid out below in italicized font and specific concerns have been numbered. Our response is given in normal font.

Reviewer #1: 

I have limited knowledge of the statistical methods used. With this limitation, my comments are:

1)I do not understand what the three study groups (with their individual controls) are: apart from Crohn's disease and ulcerative colitis, there is a third group of "IBD". What is this third group?

The author’s answer: Thank you for raising this question. We found some errors in the description of this question in the manuscript. The current study used a two-sample, multivariate Mendelian randomization analysis. Crohn's disease(CD) and ulcerative colitis(UC) are subtypes of inflammatory bowel diseases(IBD). In our study, two subgroups were analyzed, and the results were detailed in Result 3.1. Accordingly, we modified the ambiguous description, as shown in lines 194-201 of the manuscript.

2)The OR is just above 1.00 in these cases. If I understand right, this means the risk of coexistence is only slightly above expected what is expected from the general population.

The author’s answer: Thank you for your comments. As you said, the OR results in this study only indicate a low risk correlation, but the phenomenon that IBD is less clear about the specific risk factors of urolithiasis, we think the analysis results are still of certain significance.

3)The Introduction is too lengthy and simplistic. This can be made more precise and focused.

The author’s answer: Thank you for your suggestions. We have cut and modified the long or repetitive narrative statements in a targeted manner.

4)The authors use a lot of introductory phrases and adjectives, which can be deleted during editing.

The author’s answer: Thank you for your comments, and we have also made some modifications in the resubmitted manuscript.

Reviewer #2: 

1)The association of inflammatory Intestinal Bowel disease and renal involvement is well known – the most frequent renal diseases are: nephrolithiasis, tubulointerstial nephritis, glomerulonephritis and amyloidosis. As an example, PubMED for “renal lithiasis and inflammatory bowel disease” retrieved 147 results.

The author’s answer: Thank you for providing the conclusion of the correlation search on the relationship between inflammatory bowel disease and renal involvement. We included the information you provided in the introduction section of the manuscript. Lines 86 to 89 are shown in the manuscript.

2)The authors used the Mendelian Randomization Analysis. This method has been criticized as follows:

“In recent years, epidemiologists have increasingly sought to employ genetic data to identify ‘causal’ relationships between exposures of interest and various endpoints – an instrumental variable approach sometimes termed Mendelian randomization.

However, this approach is subject to all of the limitations of instrumental variable analysis and to several limitations specific to its genetic underpinnings, including confounding, weak instrument bias, pleiotropy, adaptation, and failure of replication.

Although the approach enjoys some utility in testing the etiological role of discrete biochemical pathways, like folate metabolism, this method must be treated with all the circumspection that should accompany all forms of observational epidemiology.

Going forward, we urge the elimination of randomization or causality in reports of its use and suggest that Mendelian randomization instead be termed exactly what it is – genetic instrumental variable analysis”.

(Kenneth J Mukamai, Meir J.Stampfer and Eric B.Rimm; European Journal of Epidemiology, volume 35, pages 93-97 (2020).

I suggest that this article should be evaluated also by a Statistics Expert.

The author’s answer: Thank you for supplementing the literature on the methodological limitations of Mendelian randomization studies. We thought this would provide greater credibility to our manuscript, and therefore, we cited this literature to lines 341-346 in the strengths and limitations section of the manuscript.

Thank you very much for your attention and time. Look forward to hearing from you.

Yours sincerely,

Xue-lin Jin

22-Dec., 2023

Panzhihua University, Affiliated Hospital, Anorectal

---

## [Decision Letter · Decision Letter 1]

30 Jan 2024

PONE-D-23-27679R1Risk Relationship between Inflammatory Bowel Disease and Urolithiasis: A two-sample mendelian randomization studyPLOS ONE

Dear Dr. Jin,

Thank you for submitting your manuscript to PLOS ONE. After careful consideration, we feel that it has merit but does not fully meet PLOS ONE’s publication criteria as it currently stands. Therefore, we invite you to submit a revised version of the manuscript that addresses the points raised during the review process.

We look forward to receiving your revised manuscript.

Kind regards,

Gurinder Kumar, MD

Academic Editor

PLOS ONE

Journal Requirements:

Additional Editor Comments:

Abstract: how many patient samples and how many SNPs were examined? It is unclear from this text, and seems perhaps contradictor to later in the paper where the 3 sets of controls are mentioned.

Reviewers' comments:

Reviewer's Responses to Questions

**Comments to the Author**

1. If the authors have adequately addressed your comments raised in a previous round of review and you feel that this manuscript is now acceptable for publication, you may indicate that here to bypass the “Comments to the Author” section, enter your conflict of interest statement in the “Confidential to Editor” section, and submit your "Accept" recommendation.

Reviewer #3: (No Response)

2. Is the manuscript technically sound, and do the data support the conclusions?

Reviewer #3: Yes

3. Has the statistical analysis been performed appropriately and rigorously? 

Reviewer #3: Yes

4. Have the authors made all data underlying the findings in their manuscript fully available?

Reviewer #3: Yes

5. Is the manuscript presented in an intelligible fashion and written in standard English?

Reviewer #3: No

6. Review Comments to the Author

Reviewer #3: Thank you for the opportunity to review this work.

The manuscript could benefit for an additional review by a native english speaker, but the manuscript is improved.

Abstract: how many patient samples and how many SNPs were examined? It is unclear from this text, and seems perhaps contradictor to later in the paper where the 3 sets of controls are mentioned.

Addition of section for mendelian randomisation limitations is alright.

7. PLOS authors have the option to publish the peer review history of their article (what does this mean?). If published, this will include your full peer review and any attached files.

Reviewer #3: No

---

## [Author Response · Author response to Decision Letter 1]

5 Mar 2024

Dear Editor,

We feel great thanks for your professional review work on our article again. As you are concerned, there are several problems that need to be addressed. Questions are laid out below in italicized font and specific concerns have been numbered. Our response is given in normal font.

1.Please review your reference list to ensure that it is complete and correct. If you have cited papers that have been retracted, please include the rationale for doing so in the manuscript text, or remove these references and replace them with relevant current references. Any changes to the reference list should be mentioned in the rebuttal letter that accompanies your revised manuscript. If you need to cite a retracted article, indicate the article’s retracted status in the References list and also include a citation and full reference for the retraction notice.

The author’s answer: We did our best to search the original text of the referenced articles, but we did not get accurate retraction information for the cited articles, such as being marked as "retracted". Therefore, we were only able to make changes to references that were subjectively controversial in our judgment. Thank you for your comments. If possible, we hope you can provide us with more precise suggestions on this issue. The specific modifications are as follows:

1)For references 4 (Romero V, Akpinar H, Assimos DG. Kidney stones: a global picture of prevalence, incidence, and associated risk factors. Rev Urol. 2010;12(2– 3):e86–e96.), there may have been insufficient matching of citation relevance, which has now been replaced with more convincing literature citations.

2)For references 15 (Katan MB. Apolipoprotein E isoforms, serum cholesterol, and cancer. Lancet. 1986 ;1(8479):507–508), it was replaced with “Davey Smith G, Hemani G. Mendelian randomization: genetic anchors for causal inference in epidemiological studies. Hum Mol Genet. 2014 Sep 15;23(R1):R89-98.”.

3)For references 33 (Dröge W. Free radicals in the physiological control of cell function. Physiol Rev. 2002;82(1):47–95.), we replaced it with more convincing reference “Liu Y, Sun Y, Kang J, He Z, Liu Q, Wu J, et al. Role of ROS-Induced NLRP3 Inflammasome Activation in the Formation of Calcium Oxalate Nephrolithiasis. Front Immunol. 2022 Jan 27;13:818625.”.

4)Reference 5, as an article published in 1986, may have retrieval problems because of its remote age, but this article is one of the earliest studies that can be traced back to the association between inflammatory bowel disease and kidney stones. If this reference does not meet the requirements, we may replace it or remove it.

2.Abstract: how many patient samples and how many SNPs were examined? It is unclear from this text, and seems perhaps contradictor to later in the paper where the 3 sets of controls are mentioned.

The author’s answer:

About the patient samples: The dataset used in this study included a total of four groups of relevant samples, specific single nucleotide polymorphism data were obtained from GWASs, including IBD (n=59957, 25,042 IBD cases and 34,915 controls) and its main subtypes, Crohn's disease (CD) (n=40266, 12,194 CD cases and 28,072 controls) and ulcerative colitis (UC) (n=45975, 12,366 UC cases and 33,609 controls). Summarized data on urolithiasis (n=218792, 5,347 urolithiasis cases and 213445 controls) were obtained from different GWAS studies from FINNGEN database. The specific changes are in lines 27 and 121 of the manuscript.

About SNPs: In this study, we aimed to investigate the causal relationship between inflammatory bowel disease (and its two subtypes, Crohn's disease and ulcerative colitis) and urolithiasis using a two-sample MR approach. Three sets of genetic instruments were utilized in the analysis, all drawn from publicly available genetic summary data. The first set consisted of 259 single-nucleotide polymorphisms (SNPs) robustly linked to inflammatory bowel disease (IBD), the second set was comprised of 81 SNPs associated with inflammatory bowel disease (CD) and the third set consisted of 175 SNPs associated with ulcerative colitis (UC). If necessary, we will be able to provide the corresponding raw data sets.

3.Is the manuscript presented in an intelligible fashion and written in standard English?

The author’s answer: We suggest you thoroughly copyedit your manuscript for language usage, spelling, and grammar. If you do not know anyone who can help you do this, you may wish to consider employing a professional scientific editing service.

We revised the language use, spelling, and grammar in the manuscript again as suggested by you. We also employed a professional scientific editor from Editage to assist us in revising the manuscript for publication in Plos one.

Thank you very much for your attention and time. Look forward to hearing from you.

Yours sincerely,

Xue-lin Jin

5-Mar., 2024

Panzhihua University, Affiliated Hospital, Anorectal

---

## [Decision Letter · Decision Letter 2]

19 Mar 2024

Risk relationship between inflammatory bowel disease and urolithiasis: a two-sample Mendelian randomization study

PONE-D-23-27679R2

Dear Dr. Jin,

We’re pleased to inform you that your manuscript has been judged scientifically suitable for publication and will be formally accepted for publication once it meets all outstanding technical requirements.

Kind regards,

Gurinder Kumar, MD

Academic Editor

PLOS ONE

Additional Editor Comments (optional):

Reviewers' comments:

Reviewer's Responses to Questions

**Comments to the Author**

1. If the authors have adequately addressed your comments raised in a previous round of review and you feel that this manuscript is now acceptable for publication, you may indicate that here to bypass the “Comments to the Author” section, enter your conflict of interest statement in the “Confidential to Editor” section, and submit your "Accept" recommendation.

Reviewer #4: All comments have been addressed

2. Is the manuscript technically sound, and do the data support the conclusions?

Reviewer #4: (No Response)

3. Has the statistical analysis been performed appropriately and rigorously? 

Reviewer #4: (No Response)

4. Have the authors made all data underlying the findings in their manuscript fully available?

Reviewer #4: (No Response)

5. Is the manuscript presented in an intelligible fashion and written in standard English?

Reviewer #4: (No Response)

6. Review Comments to the Author

Reviewer #4: (No Response)

7. PLOS authors have the option to publish the peer review history of their article (what does this mean?). If published, this will include your full peer review and any attached files.

Reviewer #4: No

---

## [Editor Report · Acceptance letter]

27 Mar 2024

PONE-D-23-27679R2 

PLOS ONE

Dear Dr. Jin, 

I'm pleased to inform you that your manuscript has been deemed suitable for publication in PLOS ONE. Congratulations! Your manuscript is now being handed over to our production team.

Kind regards, 

on behalf of

Dr. Gurinder Kumar 

Academic Editor

PLOS ONE